# *Listeria monocytogenes*, *Escherichia coli* and Coagulase Positive Staphylococci in Cured Raw Milk Cheese from Alentejo Region, Portugal

**DOI:** 10.3390/microorganisms11020322

**Published:** 2023-01-27

**Authors:** Joana Praça, Rosália Furtado, Anabela Coelho, Cristina Belo Correia, Vítor Borges, João Paulo Gomes, Angela Pista, Rita Batista

**Affiliations:** 1Food Microbiology Laboratory, Food and Nutrition Department, National Institute of Health Doutor Ricardo Jorge, Avenida Padre Cruz, 1649-016 Lisbon, Portugal; 2Faculty of Farmacy, University of Lisbon, Avenida Prof. Gama Pinto, 1649-003 Lisbon, Portugal; 3Genomics and Bioinformatics Unit, Department of Infectious Diseases, National Institute of Health Doutor Ricardo Jorge, Avenida Padre Cruz, 1649-016 Lisbon, Portugal; 4National Reference Laboratory for Gastrointestinal Infections, Department of Infectious Diseases, National Institute of Health Doutor Ricardo Jorge, Avenida Padre Cruz, 1649-016 Lisbon, Portugal

**Keywords:** *Listeria monocytogenes*, *Escherichia coli*, Coagulase Positive Staphylococci, whole-genome sequencing, raw milk cheese, Alentejo Portugal

## Abstract

Traditional cheeses are part of the Portuguese gastronomic identity, and raw milk of autochthonous species is a common primary ingredient. Here, we investigated the presence of *Listeria monocytogenes*, Coagulase Positive Staphylococci (CPS) and pathogenic *Escherichia coli*, as well as of indicator microorganisms (*E. coli* and other *Listeria* spp.) in 96 cured raw milk cheeses from the Alentejo region. Whole genome sequencing (WGS) of pathogenic *E. coli* and *Listeria* spp. as well as antimicrobial resistance (AMR) screening of *E. coli* isolates was also performed. *L. monocytogenes*, CPS > 10^4^ cfu/g and Extraintestinal *E. coli* were detected in 15.6%, 16.9% and 10.1% of the samples, respectively. Moreover, *L. monocytogenes* > 10^2^ cfu/g and Staphylococcal enterotoxins were detected in 4.2% and 2.2% of the samples, respectively. AMR was observed in 27.3% of the *E. coli* isolates, six of which were multidrug resistant. WGS analysis unveiled clusters of high closely related isolates for both *L. monocytogenes* and *L. innocua* (often correlating with the cheese producer). This study can indicate poor hygiene practices during milk collection/preservation or during cheese-making procedures and handling, and highlights the need of more effective prevention and control measures and of multi-sectoral WGS data integration, in order to prevent and detect foodborne bacterial outbreaks.

## 1. Introduction

As defined in European Regulation No. 853/2004, “raw milk” means “milk produced by the secretion of the mammary gland of farmed animals that has not been heated to more than 40 °C or undergone any treatment that has an equivalent effect” [1].

Due to the absence of a heat treatment, raw milk can harbor a diverse microbial flora, including both spoilage and pathogenic microorganisms, with the potential of causing human illness [2].

Data published by Eurostat concerning milk production in 2021 [3] shows that of the total raw milk (161.0 million tonnes) produced in European Union (EU) farms, the largest portion (155.2 million tonnes) was cows’ milk, the rest being ewes’ milk (3 million tonnes), goats’ milk (2.5 million tonnes) and buffalos’ milk (0.3 million tonnes). Concerning the final use of the total produced milk, only a small portion (10.4 million tonnes) was used on farms (consumed by the farmers and their families, sold directly to consumers, used as feed or processed directly), the largest portion (150.7 million tonnes) being delivered to dairies, with the predominance (146.5 million tonnes) of cows’ milk.

In 2021, the 10.4 million tonnes of cheese produced in the EU were manufactured from 16.4 million tonnes of skimmed milk together with 61.4 million tonnes of whole milk [3].

Cheese production is a process dating back several thousands of years. The earliest indication of a cheese-making process is the one found in cave paintings from around 5000 BC, representing the oldest technological application of enzymes [4]. The use of enzymes has made possible the intentional conversion of milk into cheese, making it safer and longer lasting. In Portugal, raw milk drinking is rare; however, traditional cheeses are part of Portuguese cultural and gastronomic identity [5], and to maintain loyalty to traditional production methods, as well as distinct taste in the final product given by the microflora of milk, several types of cheeses are made from unprocessed milk. Every traditional cheese originates from a complex system which results in unique organoleptic characteristics. In Portugal, some of these cheeses (*n* = 11) have obtained a registration as product with a Protected Designation of Origin (PDO), a designation of which the main purpose is to designate products that have been produced, processed and developed in a specific geographical area, using the recognized know-how of local producers and ingredients from the region concerned [6]. 

Contaminating microbes, when present in the raw milk, can persist and remain viable, even after aging the cheese for prolonged periods, which make raw milk cheeses a potential source of microbiological hazards. Raw milk cheese contamination may be related either to the contamination of raw milk or other ingredients, to pathogens from animals or the environment, or to the health and hygiene of the workers and the hygiene of the surfaces in contact with the cheese.

According to the European Union One Health 2020 Zoonoses Report [7], ‘milk’, ‘cheese’ and ‘dairy products’ were reported in 16 strong-evidence outbreaks by eight member states with 325 cases, 67 hospitalizations and one death. Various types of ‘cheese’, including soft cheese, raw milk cheese and other unspecified cheeses, were identified as the implicated vehicle in outbreaks caused by *Salmonella* Enteritidis, *Listeria monocytogenes*, Shiga toxin-producing *Escherichia coli* (STEC) and *Staphylococcus aureus* toxins. The most severe outbreak, in terms of the highest number of deaths (10 deaths), was reported by Switzerland and was associated with the consumption of cheese contaminated by *L. monocytogenes* serovar 4b [7,8].

Due to the social and economic importance that raw milk cheeses have in Portugal, the particularities of its production, and categorizing them as potential “risky” foods concretely for certain vulnerable groups of consumers, it is important for the research community to consider this a priority work area and, in this context, to evaluate the microbiological quality of this traditional foodstuff. 

The aim of this study was to evaluate the presence and enumerate some common foodborne pathogens such as *L. monocytogenes*, pathogenic *E. coli* and Coagulase Positive Staphylococci (CPS), as well as indicator microorganisms (non-pathogenic *E. coli* and *Listeria* spp. other than *L. monocytogenes*) in cured raw milk cheeses from the Alentejo region. In order to better understand the risks across the food chain, *Listeria* spp. and pathogenic *E. coli* isolates characterization by whole genome sequencing (WGS) was also performed. Since the use of antibiotics for the control of diseases in food-producing animals is a common practice in veterinary medicine and, in recent years, numerous bacteria with multiple drug resistance patterns have emerged, *E. coli* isolates (pathogenic and non-pathogenic) antimicrobial resistance (AMR) was also studied. 

## 2. Materials and Methods

### 2.1. Sampling

Ninety-six cured raw milk cheeses from different batches, corresponding to 30 brands produced by 20 identified producers located in the Alentejo region of Portugal were analyzed. The Alentejo region is one of Portugal regions with the most appreciated traditional cheeses [5] and there is a lack of studies carried out on their microbiological quality. The Alentejo Regional Coordination and Development Commission divides Alentejo region into 4 sub-regions: Alto Alentejo, Alentejo Central, Alentejo Litoral and Baixo Alentejo. Figure 1 shows the number of brands tested, by region/sub-region. Cheeses were purchased from different hypermarkets, supermarkets, local markets and grocery stores around the Lisbon region from June 2021 to May 2022. Samples were stored at refrigeration temperature (2 °C to 4 °C) from the time of purchasing until processing, within 24 h after collection, and were analyzed during their assigned shelf-life period. 

### 2.2. Microbiological Analysis

All microbiological analyses were performed according to the general requirements and guidance for microbiological examinations described in ISO 7218:2007 [9]. 

The 96 collected samples were examined for the presence of *Listeria* spp. in 25 g. *L. monocytogenes* enumeration was performed in the positive samples (result: *L. monocytogenes* detected in 25 g). Eighty-nine (89) out of the 96 samples were also tested for *E. coli* and Coagulase Positive Staphylococci (CPS) detection and enumeration. For those samples with a CPS concentration ≥4.9 × 10^4^ cfu/g, Staphylococcal enterotoxins (SE) detection was also performed.

#### 2.2.1. *E. coli* and Coagulase Positive Staphylococci (CPS) Detection and Enumeration

Each cheese sample (test portion of 25 g) was added to 225 mL of sterile Buffered peptone water (BPW-Oxoid, Basingstoke, Hampshire, UK) and homogenized at 230 rpm for 1 min using a stomacher (Stomacher, 400 Circulator, London, UK). Appropriate decimal dilutions to 10^−3^ were prepared in Tryptone salt diluent (Biokar Diagnostics, Pantin, France). Detection and enumeration of *E. coli* and Coagulase Positive Staphylococci were performed by the AFNOR validated TEMPO^®^ EC and TEMPO^®^ STA automated most probable number (MPN) system (bioMérieux, Marcyl l’Etoile, France), respectively, following the manufacturer’s instructions.

Simultaneously to TEMPO^®^ EC *E. coli* enumeration, the initial suspension 1/10 Cheese/BPW mixture was incubated at 37 °C during 24 h ± 2 h. *E. coli* plating-out was performed by streaking a loopful of this culture medium on the surface of Chromogenic Coliform Agar (CCA, Biokar Diagnostics) plates and incubated at 37 °C during 24 h± 2 h. *E. coli* colonies were selected and sub-cultured on Columbia Agar + 5% Sheep Blood (COS; bioMérieux) and incubated at 37 °C during 24 h ± 2 h, where hemolytic activity was determined. The identification of presumptive isolates was confirmed by biochemical identification on VITEK^®^2 compact system (bioMérieux). All positive isolates were stored at -80 °C in broth with 20% glycerol.

#### 2.2.2. *L. monocytogenes* and *Listeria* spp. (Not including *L. monocytogenes*) Detection and Enumeration

For *L. monocytogenes* detection, the ISO 11290-1 horizontal method [10] was followed in parallel with the alternative method-VIDAS^®^LMO2 (bioMérieux). A primary enrichment was prepared with 25 g of cheese sample in 225 mL of half-Fraser broth (bioMérieux), homogenized in a stomacher for 1 min and incubated at 30 °C during 25 h ± 1 h. One hundred microliters of the incubated suspension (primary enrichment) were transferred to 10 mL of secondary enrichment medium Fraser broth (bioMérieux) and incubated at 37 °C during 24 h ± 2 h. After incubation, 0.5 mL of the culture was tested in the VIDAS^®^ LMO2 automated system, according to the manufacturer’s instructions.

For *L. monocytogenes* enumeration, the ISO/11290-2 horizontal method [11] was followed. One milliliter of a 1:10 homogenized initial suspension (10 g of cheese + 90 mL of BPW) was spread in equal parts on the surface of three Microinstant^®^ Listeria Agar (Ottaviani e Agosti) (Biokar Diagnostics) plates and incubated at 37 °C during 48 h ± 2 h. *L. monocytogenes* presumptive colonies (blue colored surrounded by an opaque halo) were counted and subsequently isolated on Columbia Agar + 5% Sheep Blood (COS; bioMérieux) at 37 °C during 24 h ± 2 h, where hemolytic activity was determined. Biochemical identification of the isolates was performed on VITEK^®^2 compact system (bioMérieux), following the manufacturer’s instructions. All positive isolates were stored at −80 °C in Tryptone Soy Broth (TSB; Biokar Diagnostics) with 20% glycerol. 

Other *Listeria* spp. colonies (blue colonies without an opaque halo), when present, were also transferred to COS agar and identity of the isolates confirmed by biochemical identification on VITEK^®^ 2 compact system.

#### 2.2.3. Staphylococcal Enterotoxins (SE) Detection

A staphylococcal enterotoxins (SE) detection was performed in all cheese samples that presented Coagulase Positive Staphylococci levels ≥4.9 × 10^4^ cfu/g. For the detection of SE, ISO 19020:2017 [12] was followed. Briefly, 25 g of cheese (10% of the shell and 90% of the inner part) suspended in 40 mL of distilled water at 38 °C ± 2 °C were homogenized in a stomacher, for 1 min and then shaken in an VXR basic Vibrax orbital shaker (Ika^®^, Staufen, Germany) at room temperature for 30 to 60 min to allow toxin diffusion. The pH of the slurry was adjusted between 3.5 and 4.0 with HCl and centrifuged at 3130× *g* for 15 min at 4 °C. The supernatant was collected and the pH adjusted to 7.5 ± 0.1 with NaOH and centrifuged again as described above. The supernatant was concentrated on a dialysis membrane with a molecular cut-off of 6000-8000 Da (Spectrum Laboratories, Rancho Dominguez, CA, USA) against 30% (*w*/*v*) of polyethylene glycol 20,000 (Merck, Darmstadt, Germany), overnight, at 4 °C. SE detection was performed using the alternative automated method VIDAS^®^ Staph enterotoxin II (SET 2) (bioMérieux).

### 2.3. Interpretation of Microbiological Results

The criteria for the interpretation of microbiological results are listed on Table 1 and were based on the following references: Commission Regulation (EC) No 2073/2005 of 15 November 2005 on microbiological criteria for foodstuffs [13], on the Luxembourg Microbiological criteria applicable to foodstuffs [14] and on the Health Protection Agency (HPA) guidelines for assessing the microbiological safety of ready-to-eat foods placed on the market [15].

According to these criteria, one brand was classified as: Satisfactory, when the results of all the analyzed samples were classified as satisfactory;Borderline, when none of the samples were unsatisfactory and the results of at least one sample was classified as borderline;Unsatisfactory/potentially injurious to health, when at least one of the samples was classified as unsatisfactory/ potential injurious to health.

### 2.4. Pathogenic E. coli Identification and Antimicrobial Susceptibility Testing

Potential pathogenic *E. coli* isolates were identified by screening for the presence of some characteristic virulence genes (*eae*, *aggR*, *elt, estp*, and *ipaH*) by multiplex PCR (modified from Persson 2007, Boisen 2012 and Fujioka 2013 [16,17,18]) and for the presence of Shiga toxins *stx1* and *stx2* [19], as previously described [20]. An *E. coli* isolate was classified as potentially pathogenic (STEC; EPEC, Enteropathogenic *E. coli*; EAEC, Enteroaggregative *E. coli*; ETEC, Enterotoxigenic *E. coli*; EIEC, Enteroinvasive *E. coli*) when at least one of the pathotype-specific genes was detected. 

*E. coli* pathogenicity was also defined after sequence analysis (some of the presumptive non-pathogenic *E. coli* isolates were sequenced because were multidrug resistant (MDR) or presented hemolytic activity). In this case, the presence of two or more Extraintestinal pathogenic *E. coli* (ExPEC) typical virulence genes were used for this pathotype classification [21].

The Kirby–Bauer method was followed for the Antimicrobial Susceptibility Testing (AST), in 55 presumptive non-pathogenic *E. coli* isolates, following the European Committee on Antimicrobial Susceptibility Testing [22] recommendations. A panel of 18 antimicrobials were used: Trimethoprim (TMP), Tigecycline (TGC), Tetracycline (TET), Sulfamethoxazole (SMX), Ciprofloxacin (CIP), Nalidixic Acid (NAL), Meropenem (MEM), Gentamicin (GMN), Erythromycin (ERY), Chloramphenicol (CHL), Ceftriaxone (CRO), Ceftazidime (CZD), Cefoxitin (FOX), Cefotaxime (COX), Cefepime (FEP), Azithromycin (AZM), Amoxicillin-Clavulanic Acid (AMC) and Ampicillin (AMP). The results were interpreted according to the EUCAST epidemiological cut-off values (ECOFFs) [22]. An isolate was classified as multidrug-resistant (MDR) when it presented resistance to three or more antimicrobial classes.

### 2.5. Listeria spp. and E. coli Whole-Genome Sequencing, In Silico Typing and Screening of E. coli Virulence/AMR Genes

Genomic DNA was extracted from fresh cultures of all *Listeria innocua* and *L. monocytogenes*, as well as from all MDR and hemolytic *E. Coli* isolates, using the ISOLATE II Genomic DNA Kit (Bioline, London, England, UK), and quantified in the Qubit fluorometer (Invitrogen, Waltham, MA, USA) with the dsDNA HS Assay Kit (Thermo Fisher Scientific, Waltham, MA, USA), according to the manufacturer’s instructions. DNA was then prepared using the NexteraXT library preparation protocol (Illumina, San Diego, CA, USA) and then cluster generation and sequencing (2 × 150 bp) on either a MiSeq, a NextSeq 550 or NextSeq 2000 instrument (Illumina) were performed. 

Regarding *Listeria* spp and *E. coli*, we performed read quality control, trimming and de novo genome assembly with the INNUca pipeline v4.2.2 “https://github.com/B-UMMI/INNUca (accessed on 19 November 2022)” [23], using default parameters. In brief, FastQC v0.11.5 “http://www.bioinformatics.babraham.ac.uk/projects/fastqc/ (accessed on 19 November 2022)” and Trimmomatic v0.36 [24] were used for reads quality control and improvement.

For *E. coli*, sequencing reads were analyzed using the Center for Genomic Epidemiology web services “http://www.genomicepidemiology.org/services/ (accessed on 19 November 2022)” in order to identify the virulence (VirulenceFinder 2.0) and antimicrobial resistance (ResFinder 4.1) genes, and for the in silico serotyping (SerotypeFinder 2.0) and in silico Multilocus Sequence Typing (MLST) (MLST 2.0).

For *Listeria* spp., de novo genome assembly was performed with SPAdes v3.14 [25], reads were aligned with Bowtie v2.2.9 [26] and the assembly was polished with Pilon v1.23 [27], as integrated in INNUca v4.2.2 [23]. Species confirmation/contamination screening was performed with Kraken2 v2.0.7 [28]. ST determination and in silico serotyping were performed with mlst v2.18.1 “https://github.com/tseemann/mlst (accessed on 19 November 2022)” and lissero v.0.9.4 “https://github.com/MDU-PHL/LisSero (accessed on 19 November 2022)”.

Sequencing reads were deposited on the European Nucleotide Archive (ENA) under the bioprojects PRJEB31216 (*Listeria* spp.) and PRJEB54735 (*E. coli*). Accession numbers for each isolate are listed in Appendix A.

### 2.6. Core-Genome Clustering Analysis of Listeria spp. Isolates

For *L. monocytogenes*, allele-calling was performed over the polished genome assemblies with chewBBACA v2.8.5 [29] using the core-genome Multi Locus Sequence Typing (cgMLST) 1748-loci Pasteur schema [30] available at Chewie-NS website “https://chewbbaca.online (downloaded on 23 June 2022)” [31]. The cgMLST clustering analysis was performed with ReporTree v.1.0.1 “https://github.com/insapathogenomics/ReporTree (accessed on 19 November 2022)” [32] using GrapeTree (MSTreeV2 method) [33], with clusters of closely related isolates being determined and characterized at a distance thresholds of 1, 4, 7 and 15 allelic differences (ADs). A threshold of seven ADs can provide a proxy to the identification of genetic clusters with potential epidemiological concordance (i.e., “outbreaks”) [34].

For *L. innocua*, in the absence of a cgMLST schema, a core-genome alignment (enrolling polished assemblies of 20 out of 21 isolates with sequencing data) was constructed with Parsnp v.1.7.4 implemented on Harvest suite [35], using the default parameters, with exception of parameter –C, which was adjusted to 2000 in order to maximize the resolution. The core-genome SNP-based clustering analysis was performed with ReporTree v.1.0.1 “https://github.com/insapathogenomics/ReporTree (accessed on 19 November 2022)” [32] using GrapeTree (MSTreeV2 method) [33], with clusters of closely related isolates being determined and characterized at SNP thresholds of 1, 4, 7 and 15 SNPs. This core-genome SNP-based clustering analysis relied on a core-genome alignment (comprising 93% of the *L. innocua* genome size) involving a total 170 variant sites. 

Interactive phylogenetic tree visualization was conducted with GrapeTree [33].

## 3. Results

### 3.1. Microbiological Quality 

Of the 89 samples tested for all the parameters, 44 (49.4%) were classified as unsatisfactory/potentially injurious to health, 30 (33.7%) as borderline and 15 (16.9%) as satisfactory (Table 2 and Appendix A).

The classification of unsatisfactory/potentially injurious to health samples was related with diverse results, the most common the detection of *E. coli* being at a level >10^4^ cfu/g (Table 2). Most of the unsatisfactory samples (37/44, 84.0%) were also borderline regarding several other results, eight of them also containing *L. monocytogenes* (Appendix A).

Concerning the 30 borderline samples, the reason for the attributed classification was also highly variable (Table 2), the most common, once more, being related to the presence of *E. coli* > 10 cfu/g and ≤10^4^ cfu/g.

Regarding satisfactory samples, nine of them (60%) were from the same producer (producer D–brands 5, 6 and 7, Table 3). The other six were from brand two (2), brand 11 (1), brand 13 (1), brand 23 (1) and brand 25 (1) (Table 3 and Appendix A). 

Considering the microbiological quality by brand/producer, 24 out of the 30 evaluated brands were classified as unsatisfactory/potentially injurious to health (24/30, 80.0%) and two as borderline (2/30, 6.7%). Only four (4/30, 13.3%) brands presented satisfactory microbiological results (5, 6, 7 and 25) and only producer D (1/20, 5%) showed satisfactory results regarding the microbiological quality for all of its brands (Table 3). 

Considering the microbiological contamination of the cheeses, it is important to highlight that in 15/96 (15.6%) of the analyzed samples, it was possible to detect *L. monocytogenes* in 25 g, of which four revealed concentrations of >10^2^ cfu/g (4/96–4.2%). Furthermore, in 24/96 (25.0%) of the samples, *Listeria* of other species (not *L. monocytogenes*) was detected, as well as 21 *L. innocua*, two *Listeria ivanovii* and one *Listeria seeligeri*. Also, 70/89 (78.7%) of the samples contained *E. coli* > 10 cfu/g, 26 (29.2%) in concentrations >10^4^ cfu/g, and in nine pathogenics, *E. coli* was isolated (9/89: 10.1%); 50/89 (56.2%) of the samples contained Coagulase Positive Staphylococci, of which 15 in levels >10^4^ cfu/g (15/89–16.9%), and in two of these samples, staphylococcal enterotoxins were detected (2/89–2.2%) (Table 3).

### 3.2. Pathogenic E. coli Identification and Antimicrobial Susceptibility Testing

None of the 89 cheeses tested for *E. coli* were considered pathogenic based on PCR for the tested virulence genes (*eae*, *aggR*, *elt*, *estp*, *invE*, *stx1* and *stx2*).

Antimicrobial susceptibility testing (AST) was performed in all hemolytic *E. coli* (*n* = 3) and in a subset of presumptive non-pathogenic *E. coli* isolates (*n* = 52), and in a total of 55 isolates. Fifteen of the 55 (15/55, 27.3%) isolates were resistant to at least one of the 18 tested antimicrobials, six of which were classified as MDR (Table 3, Table 4 and Appendix A). 

Genomic analysis of hemolytic (*n* = 3) and MDR (*n* = 6) *E. coli* isolates showed that all of them were classified as ExPEC (Table 3,Table 4 and Appendix A). 

Ten of the 15 *E. coli* AMR isolates were detected in cheese samples also containing *L. monocytogenes* and/or Coagulase Positive Staphylococci. In the case of Coagulase Positive Staphylococci, some of these samples contained concentrations > 10^4^ cfu/g and in one sample, it was possible to identify the enterotoxin producer staphylococci (Table 4). 

### 3.3. E. coli and Listeria Monocytogenes Typing

Amongst the ExPEC isolates, six serotypes and Sequence Types (ST) were identified: three isolates were identified as O8:H25 and belonged to ST58 (brands 21, 29 and 30); two isolates were O1:H32 and ST10 (brands 21 and 26); one isolate was O15:H18 and ST69 (brand 18); one isolate was O4:H16 and ST1145 (brand 4); one isolate was O142:H38 and ST154 (brand 17); and the other was O6:H11 and ST73 (brand 16) (Appendix A).

Among the 15 *L. monocytogenes* isolates, six ST were identified: eight isolates were identified as belonging to ST788 (six isolated from brands 17, 18, 20 and 21, from producer L, one from brand 23, producer N, and one from brand 24, producer O); three as ST378 (two from brand 3, producer B, and one from brand 17, producer L); one as ST1 (from brand 19, producer L); one as ST9 (from brand 19, producer L); one as ST666 (from brand 24, producer O); and one as ST87 (from brand 28, producer R) (Appendix A).

### 3.4. Core-Genome Clustering Analysis of Listeria spp. Isolates

In order to assess the genetic relatedness among *L. monocytogenes* and *L. innocua* food isolates, and its correlation with cheese producer/brands, a core-genome clustering analysis was performed (Figure 2). For *L. monocytogenes*, the cgMLST analysis (comprising 15 isolates) revealed two genetic clusters of high closely related isolates (≤ 7 ADs): cluster A (enrolling isolates 12 and 13, both from producer B) and cluster B (enrolling 6 isolates—1, 4, 6, 7, 14 and 15 from producer L, and the isolate 3 from producer N) (Figure 2A; Appendix A). Of note, we found more than one *L. monocytogenes* strain in cheeses from the same producer, namely two strains (belonging to sequence types ST788 and ST666) from producer O and four strains (belonging to sequence types ST1, ST9, ST378 and ST788) from producer L. For *L. innocua* (20 sequenced isolates, all belonging to ST1085), the genetic clustering perfectly correlated with the producer (Figure 2B; Appendix A), with same-producer isolates being interconnected by ≤ 12 SNPs. Notably, similarly to *L. monocytogenes*, most *L. innocua* isolates were linked to producer L. 

## 4. Discussion

According to the EU’s data from monitoring foodborne outbreaks, between 2015 and 2020, several outbreaks were associated with cheese consumption: five were caused by *L. monocytogenes*, with 47 human cases, 43 hospitalizations and 11 deaths; 73 were caused by *S. aureus* toxins, with 1040 human cases, 108 hospitalizations and no deaths; and 8 were caused by STEC, with 53 human cases, 24 hospitalizations and 2 deaths [36]. These data show that *L. monocytogenes*, *S. aureus* and pathogenic *E. coli* are bacteria capable of surviving, multiplying and/or producing toxins throughout different stages in farm, production (cheese-making process) and consumer levels, constituting a microbiological risk and potentially causing disease after cheese consumption. 

In fact, in this study, all these pathogens were often found in the 96 analyzed cheeses and, in the majority of them, in concentrations that classified them as unsatisfactory/potentially injurious to health or borderline, from a microbiological point of view.

The prevalence of *L. monocytogenes* found by other authors in cheese samples around Europe are diverse. Most of the studies reported the prevalence of *Listeria monocytogenes* as lower than the one found in this study: Little et al. [37] analyzed 1819 raw milk cheeses, from United Kingdom (UK), and detected *Listeria monocytogenes* in 17 (0.9%), one in concentrations above 100 cfu/g; O’Brien et al. [38] studied 351 cheeses from 15 Irish producers, and reported a prevalence of *L. monocytogenes* of 6%; Rudol et al. [39] reported a prevalence of 6.4% of *L. monocytogenes* after analyzing 329 European red smear cheese samples; and Almeida et al. [40] examined 70 raw milk Portuguese cheeses, and encountered *L. monocytogenes* in 8 (11.4%), one in concentrations of >100 cfu/g. Moreover, in some cases, authors could not detect *L. monocytogenes* in the tested cheese samples [41,42,43,44]. However, there is at least one European study that reported a value of the prevalence of *L. monocytogenes* similar to the one found in this work (15.6%); Coroneo et al. [45] tested 87 samples of Ricotta Salata, produced in Sardinia, and stated that 17.2% of the samples were positive for the presence of *L. monocytogenes*. Also, in accordance with the results found in this work, other authors also reported the presence of *Listeria* species, other than *L. monocytogenes*, in the evaluated cheese samples [39,46]. These species, although not considered pathogenic, are important indicators of the possible presence of *L. monocytogenes*, and its presence should be considered. 

Recently, cases of listeriosis are increasingly at a multinational level and are frequently related to the consumption of cheeses [47]. In EU, at least five recent listeriosis outbreaks were correlated with the consumption of this foodstuff: a commercial cheese (acid curd) made from pasteurized milk in Germany, in 2006–2007 [48]; a quargel cheese in Austria, Germany and Czech Republic in 2009–2010 [49]; a hard cheese made with pasteurized milk in Belgium in 2011 [50]; a Latin-style fresh cheese made from pasteurized milk in Spain in 2012 [51] and a fresh cheese made from pasteurized cow and goat milk in Portugal in 2009–2012 [52].

Similar to *L. monocytogenes*, the prevalence values of *S. aureus* and *E. coli* found in cheese samples around Europe are also divergent, and are sometimes difficult to compare due to the distinct cut-off values applied among studies. Giammanco et al. [44] analyzed 50 Pecorino Siciliano (PS) “primosale” cheeses in Italy and reported a prevalence of *S. aureus* coagulase positive in concentrations >10^5^ cfu/g of 4% and of *E. coli* ≥ 10^3^ cfu/g of 44%; Little et al. [37] in the UK detected *S. aureus* > 10^4^ cfu/g in 13/1819 (0.7%) of the analyzed raw milk cheeses and *E. coli* ≥ 10^3^ cfu/g in 1.4% of the samples; Almeida et al. [40] in Portugal identified *S. aureus* > 10^4^ cfu/g in 5.7% and *E. coli* > 10^4^ cfu/g in 21.4% of the samples and Rosengren et al. [42] in Sweden described *S. aureus* >10^5^ cfu/g in 10.9% and *E. coli* ≥10^5^ cfu/g in 3.6% of the samples. Moreover, in accordance with the results presented in this study, several studies more focused on the detection of *S. aureus* in cheeses reported not only high prevalence values of this microorganism but also the presence of staphylococcal enterotoxins [43,53].

The prevalence values encountered in this study, as well as the ones reported in other studies around Europe, clearly demonstrate that *E. coli*, *S. aureus* and *Listeria* spp. are microorganisms that are frequently detected in raw milk cheeses and are sometimes present in concentrations above the normative levels, and consequently may potentially cause disease. 

Although several studies in Europe have already described the presence of STEC isolates in cheese samples [54,55,56,57], in this study, we did not find this pathotype in the tested samples. However, nine ExPEC strains were isolated. It is important to notice that many ExPEC strains found in humans with urinary tract infection, sepsis and other extraintestinal infections, particularly the most resistant to antimicrobials, may have a food animal source and may be transmitted via the food supply [58]. In fact, six out of the nine identified ExPEC isolates were MDR. Moreover, the detection of *E. coli* isolates resistant to antimicrobials in 15 cheeses, and the concomitant presence of at least one of the other tested microorganisms in ten of them, highlights the potential horizontal transfer of antibiotic resistance genes among these cohabiting bacteria and also, eventually, to other gut bacteria, through cheese consumption. Bacterial antibiotic resistance, in particular MDR, has become a global challenge, threatening human and animal health [59]. It is estimated that by 2050, the number of deaths accounted for by MDR will be higher than the ones due to cancer [60].

Moreover, the six MDR ExPEC isolates belonging to three STs (ST10, ST58, ST69) are already associated to human disease. The *E. coli* ST10 clonal complex is among the emerging ExPEC lineages. Although commonly encountered as an antimicrobial-susceptible low-virulence human intestinal colonizer, it has also been associated with human infections [58]. *E. coli* ST58 has emerged as a prominent sequence type and a globally disseminated uropathogen that often progresses to sepsis [61]. *E. coli* ST69 accounted for 4% of the *E. coli* isolates causing extraintestinal infections in Spain, two of them being also characterized as belonging to O15:H18 serotype, the one detected in our study [62].

Regarding *L. monocytogenes* typing, ST1, ST9, and ST87 clonal complexes, found in this study in five *L. monocytogenes* isolates, were already reported in human clinical isolates in at least one of two large WGS studies regarding the characterization of *L. monocytogenes* isolates in foodstuffs and human samples [30,63]. ST1 and ST87 were also the two most frequent sequence types reported in a study performed in Gipuzkoa in Northern Spain, aiming to describe the clinical features and the molecular epidemiology of human listeriosis over the 2010–2020 period [64]. 

WGS techniques, when combined with epidemiological information, have the potential to attribute relatedness among studied strains and thus to establish links between human disease cases and causative suspect food vehicles. Regarding the cgMLST analysis of the 15 *L. monocytogenes* isolates, it is noteworthy that the detection of genetic clusters of high closely related isolates (one of them involving two producers), as well as the identification of highly genetically distant strains, were linked to the same producer(s) (Figure 2A). These results suggest that *L. monocytogenes* cheese contamination may be related with bad manufacturing and hygienic practices during cheese production or transportation, since all these cheeses were purchased in different locations and belong to different batches. The identification of one isolate from producer N in Cluster B may be justified by the fact that producers L and N were located on the same street and may share suppliers, distribution chain, etc. When integrating the cgMLST results of the *L. monocytogenes* isolates found in this study, in the global WGS *L. monocytogenes* collection of the National Institute of Health database, it was possible to verify that three of the *L. monocytogenes* cheese isolates potentially matched with clinical isolates from 2009 to 2022 (data not shown). These results suggest a potential relatedness among these *L. monocytogenes* strains, the cheeses from which they were isolated and the reported human listeriosis cases. These results were communicated to the relevant Portuguese authorities and are subsequently under investigation.

The core-genome SNP-analysis of *L. innocua* isolates reinforces the idea that *Listeria* spp. cheese contamination is related to bad manufacturing and hygienic practices during cheese production or transportation. Three different clusters were detected, all of them producer-specific (Figure 2B).

The results revealed that in Alentejo’s cheese factories, the investment in training in food safety procedures should be reinforced and the analysis for microbial control are not sufficient or not carried out with the desirable periodicity.

## 5. Conclusions

In conclusion, despite the existence of European regulation applicable to raw milk cheeses during the production process and when placed in the market, the contamination detected in a significant number of the cheese samples analyzed within our study alerts for the need of improving the compliance with the good manufacturing and hygienic practices along the different levels of the food chain (farm, artisanal production and consumer). 

Considering the possible exposure of the consumer to the above-mentioned pathogenic microorganisms in dairy products made from raw milk, appropriate risk communication on the consumption of these products, particularly to vulnerable populations, is recommended.

It is also crucial to develop enhanced strategies, controlling the initial microbial load and the presence of pathogenic microorganisms in raw milk and the dairy farm environment, therefore monitoring potential hazards along the manufacturing of artisanal cheeses in order to contribute to the prevention of foodborne diseases involving these types of traditional Portuguese products. In addition, this study shows the need for a systematic integration of genomic data at a multi-sectorial level towards an enhanced routine surveillance and outbreak investigation of foodborne diseases.

## Figures and Tables

**Figure 1 microorganisms-11-00322-f001:**
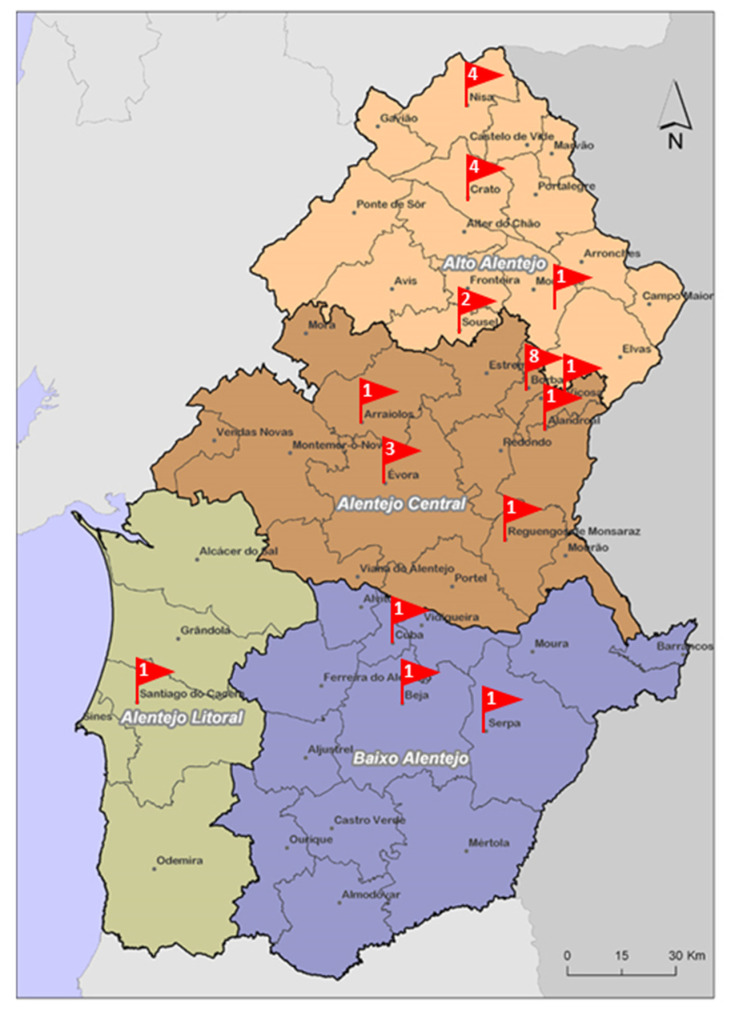
Number of brands assessed per Alentejo region.

**Figure 2 microorganisms-11-00322-f002:**
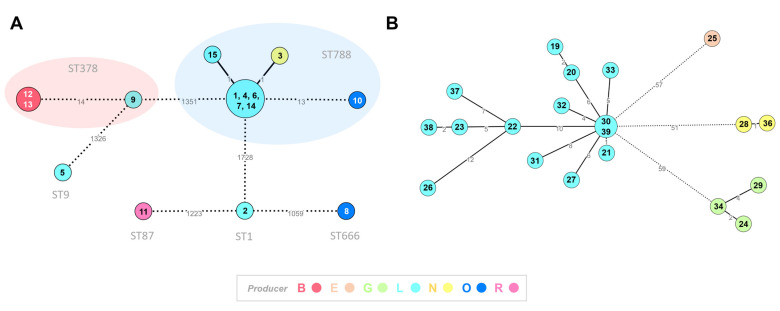
Core-genome clustering analysis of *Listeria* spp. isolates. (**A**) For *L. monocytogenes* (15 isolates), the Minimum Spanning Tree (MST) was constructed based on the cgMLST 1748-loci Pasteur schema [30]. Each circle (node) contains the strain’s designation and represents a unique allelic profile, with numbers on the connecting lines representing allelic distances (AD) between nodes. Straight and dotted lines reflect nodes linked with ADs below and above a threshold of seven ADs, which can provide a proxy to the identification of genetic clusters with potential epidemiological concordance [34]. The traditional seven-loci MLST classification is also indicated. (**B**) For *L. innocua* (20 isolates), the MST was constructed based on a core-genome SNP-based alignment (comprising 93% of the *L. innocua* genome size) involving a total 170 variant sites. Each circle (node) contains the strain’s designation and represents a unique SNP profile, with numbers on the connecting lines representing SNP distances between nodes. Straight and dotted lines reflect nodes linked with a SNP distance below and above a threshold of 15 SNPs. For both panels, data visualization was adapted from GrapeTree dashboard [33], with the node colors reflecting the producer.

**Table 1 microorganisms-11-00322-t001:** Criteria for the interpretation of microbiological results according to the selected parameters.

		Interpretation		
Parameters		Satisfactory	Borderline	Unsatisfactory/Potential Injurious to Health
**Pathogens**	* **L. monocytogenes** *	Not detected in 25 g	≤10^2^ cfu/g	>10^2^ cfu/g
	**CPS**	<10 cfu/g	10–≤10^4^ cfu/g	>10^4^ cfu/g
	**STEC and** **non-STEC**	Not detected in 25 g	N/A	Detected in 25 g
**Indicator organisms**	* **E. coli** *	<10 cfu/g	10–≤10^4^ cfu/g	>10^4^ cfu/g
	***Listeria*****spp. (not*****L. monocytogenes***)	Not detectedin 25 g	≤10^2^ cfu/g	>10^2^ cfu/g

**CPS**—Coagulase Positive Staphylococci; **N/A**—Not Applicable; **STEC**—Shiga toxin-producing *E. coli*; **cfu/g**—colony-forming units per gram.

**Table 2 microorganisms-11-00322-t002:** Interpretation of Unsatisfactory and Borderline results in the 89 samples tested for all the criteria.

**MQ**	**Results**	**# Samples**
**Unsatisfactory**	*E. coli* > 10^4^ cfu/g	21 (4 also ExPEC)
	CPS > 10^4^ cfu/g	11 (1 SE+)
	*L. monocytogenes* > 100 cfu/g	3
	ExPEC	4
	*E. coli* and CPS > 10^4^ cfu/g	4 (1SE+ and ExPEC)
	*L. monocytogenes* > 100 cfu/g and *E. coli* > 10^4^ cfu/g	1
	**Total**	**44**
	**Results**	**# Samples**
**Borderline**	*E. coli* > 10 and ≤10^4^ cfu/g	9
	CPS > 10 and ≤10^4^ cfu/g	2
	*Listeria* ssp. ≠ *L. monocytogenes* Detected in 25 g and <100 cfu/g	1
	*E. coli* and CPS > 10 and ≤10^4^ cfu/g	8
	*Listeria* spp. other than *L. monocytogenes* Detected in 25 g and <100 cfu/g; *E. coli* > 10 and ≤10^4^ cfu/g	3
	*L. monocytogenes* Detected in 25 g and <100 cfu/g; *E. coli* > 10 and ≤10^4^ cfu/g	2
	*Listeria* spp. ≠ *L. monocytogenes* Detected in 25 g and <100 cfu/g; *E. coli* and CPS > 10 and ≤10^4^ cfu/g	4
	*L. monocytogenes* Detected in 25g and <100 cfu/g; *Listeria* spp. ≠ *L. monocytogenes* Detected in 25 g and <100 cfu/g; *E. coli* and CPS > 10 and ≤10^4^ cfu/g	1
	**Total**	**30**

**#**—**Number; MQ**—Microbiological quality; **CPS**—Coagulase Positive Staphylococci; **cfu/g**—colony-forming units per gram, **SE**—Staphylococcal enterotoxins; **ExPEC**—Extraintestinal *E. coli.*

**Table 3 microorganisms-11-00322-t003:** Microbiological results by brand and producer.

Prod	Brand-Milk Type	Region	# Samples *	*L. monocytogenes*	Other *Listeria* spp.		*E. coli*				CPS			MQ
				D/25 g	>10^2^ cfu/g	D/25 g	>10 <10^4^ cfu/g	>10^4^ cfu/g	Pathogenic	AMR	>10 <10^4^ cfu/g	>10^4^ cfu/g	Toxin	
A	1-Ewe (with chili)	Alto Alentejo	3/1/1	0	0	0	0	**1**	0	0	0	0	0	U
	2-Ewe		4/4/1	0	0	0	2	0	0	0	2	0	0	B
B	3-Ewe		5/5/2	**2**	0	0	4	**1**	0	1	0	0	0	U
C	4-Ewe, cow and goat		3/3/3	0	0	0	1	**2**	**1 (ExPEC)**	0	2	0	0	U
D	5-Ewe		3/3/0	0	0	0	0	0	0	0	0	0	0	S
	6-Ewe (PDO)		3/3/0	0	0	0	0	0	0	0	0	0	0	S
	7-Ewe and goat		3/3/0	0	0	0	0	0	0	0	0	0	0	S
E	8-Ewe, cow and goat		3/3/2	0	0	1	2	**1**	0	0	0	**2**	0	U
F	9-Ewe		3/3/3	0	0	2	0	**3**	0	0	2	**1**	0	U
G	10-Ewe		4/4/3	0	0	3	4	0	0	0	2	**1**	0	U
H	11-Ewe (PDO)		4/3/2	0	0	0	2	0	0	0	0	**1**	0	U
I	12-Ewe and cow	Alentejo Central	5/5/3	0	0	0	3	0	0	0	3	**1**	**1**	U
	13-Ewe (PDO)		3/3/0	0	0	0	2	0	0	0	2	0	0	B
	14-Ewe and cow		3/3/2	0	0	0	3	0	0	1	0	**3**	0	U
J	15-Ewe		2/2/2	0	0	0	1	**1**	0	1	1	0	0	U
K	16-Ewe		2/2/1	0	0	0	2	0	**1 (ExPEC)**	0	1	0	0	U
L	17-Ewe and cow		4/3/2	**2**	**1 (10^3^)**	2	3	0	**1 (ExPEC)**	1	1	**1**	0	U
	18-Ewe and cow		7/6/5	**1**	0	5	4	**1**	**1 (ExPEC)**	2	2	0	0	U
	19-Ewe and cow		3/3/3	**2**	0	2	3	0	0	0	1	**2**	0	U
	20-Ewe		3/3/3	**2**	**1 (10^3^)**	3	0	**3**	0	1	3	0	0	U
	21-Ewe		3/3/3	**2**	0	3	0	**3**	**2 (ExPEC)**	2	2	**1**	**1**	U
M	22-Ewe and cow		4/3/3	0	0	0	0	**3**	0	0	2	**1**	0	U
N	23-Ewe and cow		3/3/2	**1**	0	2	1	**1**	0	0	2	0	0	U
O	24-Ewe		3/3/1	**2**	**2 (10^2^; 10^3^)**	0	2	0	0	1	2	0	0	U
	25-Ewe (PDO)		1/1/0	0	0	0	0	0	0	0	0	0	0	S
P	26-Ewe and cow		2/2/2	0	0	0	2	0	**1 (ExPEC)**	1	0	0	0	U
Q	27-Ewe	Alentejo Litoral	2/1/1	0	0	0	0	**1**	0	0	0	0	0	U
R	28-Ewe (PDO)	Baixo Alentejo	3/3/2	**1**	0	1	1	**2**	0	1	2	**1**	0	U
S	29-Ewe		2/2/1	0	0	0	1	**1**	**1 (ExPEC)**	1	0	0	0	U
T	30-Ewe (PDO)		3/3/2	0	0	0	1	**2**	**1 (ExPEC)**	2	3	0	0	U
		**Total**	96/89/55	15/96 (15.6%)	4/96 (4.2%)	24/96 (25%)	44/89 (49.4%)	26/89 (29.2%)	9/89 (10.1%)	15/55 (27.3%)	35/89 (39.3%)	15/89 (16.9%)	2/89 (2.2%)	

**#**: **Number; Prod**: Producer; **D**: Detected; *: samples tested for *Listeria* spp. detection/ samples tested for *Listeria* spp. detection and *E. coli* and *Staphylococcus* enumeration/samples tested for *E. coli* pathogenicity and antimicrobial resistance (AMR); **CPS**: Coagulase Positive Staphylococci; **PDO**: Protected Designation of Origin; **ExPEC**: Extraintestinal *E. coli*; **cfu/g**: colony-forming unit per gram; **MQ**: Microbiological quality; **U**: Unsatisfactory/potentially injurious to health; **B**: Borderline; **S**: Satisfactory.

**Table 4 microorganisms-11-00322-t004:** AMR *E. coli* and the concomitant presence of *L. monocytogenes* and Coagulase Positive Staphylococci.

	*E. coli*								*Listeria monocytogenes* cfu/g	Coagulase Positive Staphylococci cfu/g
	AMR						cfu/g	Pathogenic (ST)		
Brand	AMP	AMC	CHL	TET	TMP	SMX				
**3**	**x**			**x**			**++**	**N**	**−**	**−**
**14**				**x**			**+**	**N**	**−**	**++**
**15**					**x**		**+**	**N**	**−**	**−**
**17**				**x**		**x**	**+**	**N**	**+**	**++**
**18**					**x**	**x**	**+**	**N**	**−**	**−**
**18 ***	**x**			**x**	**x**	**x**	**+**	**ExPEC (ST69)**	**+**	**+**
**20**				**x**	**x**	**x**	**++**	**N**	**+**	**+**
**21 ***	**x**			**x**	**x**		**++**	**ExPEC (ST58)**	**−**	**++ SE**
**21 ***	**x**	**x**	**x**	**x**		**x**	**++**	**ExPEC (ST10)**	**+**	**+**
**24**	**x**				**x**	**x**	**+**	**N**	**+**	**+**
**26 ***	**x**		**x**	**x**		**x**	**+**	**ExPEC (ST10)**	**−**	**−**
**28**				**x**		**x**	**++**	**N**	**+**	**++**
**29 ***	**x**			**x**	**x**		**++**	**ExPEC (ST58)**	**−**	**−**
**30 ***	**x**			**x**	**x**		**++**	**ExPEC (ST58)**	**−**	**+**
**30**	**x**				**x**	**x**	**+**	**N**	**−**	**+**

**AMR**: Antimicrobial resistance; **AMP**: Ampicillin; **AMC**: Amoxicillin-Clavulanic Acid; **CHL**: Chloramphenicol; **TET**: Tetracycline; **TMP**: Trimethoprim; **SMX**: Sulfamethoxazole; **N**: no; **x**—present; **+** means “present in 25g” for *Listeria monocytogenes* and “>10 and ≤10^4^” for *E. coli* and Coagulase positive *Staphylococcus*; ***++*** means >10^2^ cfu/g for *Listeria monocytogenes* and >10^4^ cfu/g for *E. coli* and Coagulase positive *Staphylococcus;*
**SE**: Staphylococcal enterotoxin; **cfu/g**: colony-forming unit per gram; *****—Classified as Multi Drug Resistant (MDR); **ST**—Sequence type.

## Data Availability

All supporting data and protocols have been provided within the article or through Appendix A. Appendix A is available with the online version of this article.

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
