# Peer review of "Listeria monocytogenes, Escherichia coli and Coagulase Positive Staphylococci in Cured Raw Milk Cheese from Alentejo Region, Portugal"

_microorganisms, 2023, doi:10.3390/microorganisms11020322_

Round 1
Reviewer 1 Report
The review work is concerned with assessing the presence of Listeria monocytogenes, Coagulase Positive Staphylococci (CPS), and pathogenic Escherichia coli, as well as indicator microorganisms (E. coli and other Listeria spp.) in 96 cured raw milk cheeses from Alentejo 24 region in Portugal. The work seems to meet the requirements for a scientific paper, and the theme presented is described in a comprehensive manner. It is worth highlighting that the results obtained have only local significance and seem incomplete.
Comments:
Material and Methods
1. The results appear to be run out. Why didn't the authors test antibiotic resistance for the other isolates (e.g.L.monocytogenes)?
2. Tables are graphically incorrect and thus unreadable, and should be corrected.
3. A separate paragraph Conclusion should be created.
Author Response
Reviewer 1
We deeply appreciate your corrections, comments, and suggestions, which will improve the quality of our manuscript. Thank you for that.
Below you will find our explanations and considerations on your revision:
Comments:
Material and Methods
- The results appear to be run out. Why didn't the authors test antibiotic resistance for the other isolates (e.g.L.monocytogenes)?
Answer: The reviewer is right when says that it would be interesting to test antibiotic resistance of all isolates. However, we have decided to perform this assay only in E. coli because this microorganism is considered an indicator for monitoring the occurrence and levels of AMR.
- Tables are graphically incorrect and thus unreadable, and should be corrected.
Answer: Thank you for your comment. We have tried to improve the readability of all the tables in the manuscript. Regarding Table 3, we have changed the table format from portrait to landscape allowing to have a one page table.
- A separate paragraph Conclusion should be created.
Answer: Thank you very much for your comment. Done.
Reviewer 2 Report
Praca and others investigated the pathogenic and/or spoilage bacteria in cured raw milk cheese from Alentejo region. The manuscript is well written and structured, but some revision is required.
Why did the authors test raw milk cheese produced in the Alentejo region only? Is this region most consuming the cheese? Please describe the reason(s) for a selecting specific region in the Introduction, Materials & methods, or Discussion sections.
The last paragraph of the Introduction section can be improved. To enhance the novelty of this manuscript, the authors emphasize WGS approach as described in the Discussion section.
In the last paragraph of the Discussion section, the authors should provide the situation of manufacturing in the dairy industry in Portugal, emerging or present controlling methods of pathogenic microorganisms in cheese making, etc. to emphasize the importance of this manuscript.
In addition, the authors need to revise the manuscript carefully because there are many errors, typos, italics of bacterial names, misuse of abbreviations, etc.
Minor comments
1) Lines 45-55: Please use “dot” instead of “comma” as the dot is for the decimal marker according to the SI units. All the units in this journal should use SI units (please refer to the instruction for authors).
2) Lines 383-384: It is better to describe the percent of prevalence for L. monocytogenes in each cheese product made from pasteurized and unpasteurized milk.
3) Figure 2: Please indicate the strain names (such as ST378, ST788, etc.) accurately though the information was described in the Results section.
Author Response
We deeply appreciate your corrections, comments, and suggestions, which will improve the quality of our manuscript. Thank you for that.
Below you will find our explanations and considerations on your revision:
Why did the authors test raw milk cheese produced in the Alentejo region only? Is this region most consuming the cheese? Please describe the reason(s) for a selecting specific region in the Introduction, Materials & methods, or Discussion sections.
Answer: We have tried to fulfil the reviewer suggestion (lines 104 and 105).
The last paragraph of the Introduction section can be improved. To enhance the novelty of this manuscript, the authors emphasize WGS approach as described in the Discussion section.
Answer: We have tried to fulfil the reviewer suggestion (line 94)
In the last paragraph of the Discussion section, the authors should provide the situation of manufacturing in the dairy industry in Portugal, emerging or present controlling methods of pathogenic microorganisms in cheese making, etc. to emphasize the importance of this manuscript.~
Answer: We have tried to fulfil this reviewer suggestion (lines 483-485)
In addition, the authors need to revise the manuscript carefully because there are many errors, typos, italics of bacterial names, misuse of abbreviations, etc.
Answer: Thank you for your comment. We have tried to check the entire manuscript and correct it according to the reviewer suggestion.
Minor comments
- Lines 45-55: Please use “dot” instead of “comma” as the dot is for the decimal marker according to the SI units. All the units in this journal should use SI units (please refer to the instruction for authors).
Answer: The reviewer is right. We are sorry for this mistake. We have corrected this part of the manuscript.
- Lines 383-384: It is better to describe the percent of prevalence for monocytogenesin each cheese product made from pasteurized and unpasteurized milk.
Answer: O’Brien et al have studied 351 cheeses. However, they didn’t specified from which cheeses (pasteurized or unpasteurized) they isolated Listeria monocytogenes. In order to avoid any misunderstanding we have changed the respective sentence in the manuscript (line 392).
- Figure 2: Please indicate the strain names (such as ST378, ST788, etc.) accurately though the information was described in the Results section.
Answer: ST378, ST788 are not strains names, are the sequence types of the isolates. We have tried to clarify this in lines 355 and 356.
Round 2
Reviewer 1 Report
The manuscript was corrected according to suggestion and can be published in its present form. I recommend author proof and confirmation.
Author Response
Thank you very much for your revision.
Reviewer 2 Report
The authors addressed most comments adequately. However, two answers are somewhat lacking.
Comment for the reason(s) for selecting a specific region
There is no reference(s) explaining the sentence “Alentejo region is one of Portugal regions with the most appreciated traditional cheeses”. Please add the reference(s).
Comment for Figure 2
First, I am sorry for misunderstanding the sequence types of the isolates.
However, I still think that Figure 2A can be improved. I cannot recognize which strains are belonging to the sequence types in Figure 2A without reading the Results section. The authors should try to revise Figure 2A to clearly indicate which strains belong to the sequence types.
Author Response
The authors appreciated the comments.
Comment for the reason(s) for selecting a specific region
There is no reference(s) explaining the sentence “Alentejo region is one of Portugal regions with the most appreciated traditional cheeses”. Please add the reference(s).
Answer: We have followed the reviewer suggestion introducing reference [5] in this part of the manuscript (line 105). The selection of this region was based on the absence of studies about the microbiological quality of this specific type of cheese.
Comment for Figure 2
First, I am sorry for misunderstanding the sequence types of the isolates.
However, I still think that Figure 2A can be improved. I cannot recognize which strains are belonging to the sequence types in Figure 2A without reading the Results section. The authors should try to revise Figure 2A to clearly indicate which strains belong to the sequence types.
Answer: We have tried to enhance the highlighted area regarding Sequence Type (ST) identification on Figure 2A. Please see if it is now clearer.